# Deep Predictive Coding Networks for Video Prediction and Unsupervised Learning

**William Lotter, Gabriel Kreiman & David Cox**
Harvard University
Cambridge, MA 02215, USA
`{lotter,davidcox}@fas.harvard.edu`
`gabriel.kreiman@tch.harvard.edu`

## Abstract

While great strides have been made in using deep learning algorithms to solve supervised learning tasks, the problem of unsupervised learning — leveraging unlabeled examples to learn about the structure of a domain — remains a difficult unsolved challenge. Here, we explore prediction of future frames in a video sequence as an unsupervised learning rule for learning about the structure of the visual world. We describe a predictive neural network ("PredNet") architecture that is inspired by the concept of "predictive coding" from the neuroscience literature. These networks learn to predict future frames in a video sequence, with each layer in the network making local predictions and only forwarding deviations from those predictions to subsequent network layers. We show that these networks are able to robustly learn to predict the movement of synthetic (rendered) objects, and that in doing so, the networks learn internal representations that are useful for decoding latent object parameters (e.g. pose) that support object recognition with fewer training views. We also show that these networks can scale to complex natural image streams (car-mounted camera videos), capturing key aspects of both egocentric movement and the movement of objects in the visual scene, and the representation learned in this setting is useful for estimating the steering angle. Altogether, these results suggest that prediction represents a powerful framework for unsupervised learning, allowing for implicit learning of object and scene structure.

## 1 Introduction

Many of the most successful current deep learning architectures for vision rely on supervised learning from large sets of labeled training images. While the performance of these networks is undoubtedly impressive, reliance on such large numbers of training examples limits the utility of deep learning in many domains where such datasets are not available. Furthermore, the need for large numbers of labeled examples stands at odds with human visual learning, where one or a few views of an object is often all that is needed to enable robust recognition of that object across a wide range of different views, lightings and contexts. The development of a representation that facilitates such abilities, especially in an unsupervised way, is a largely unsolved problem.

In addition, while computer vision models are typically trained using static images, in the real world, visual objects are rarely experienced as disjoint snapshots. Instead, the visual world is alive with movement, driven both by self-motion of the viewer and the movement of objects within the scene. Many have suggested that temporal experience with objects as they move and undergo transformations can serve as an important signal for learning about the structure of objects (Földiák, 1991; Softky, 1996; Wiskott & Sejnowski, 2002; George & Hawkins, 2005; Palm, 2012; O'Reilly et al., 2014; Agrawal et al., 2015; Goroshin et al., 2015a; Lotter et al., 2015; Mathieu et al., 2016; Srivastava et al., 2015; Wang & Gupta, 2015; Whitney et al., 2016). For instance, Wiskott and Sejnowski proposed "slow feature analysis" as a framework for exploiting temporal structure in video streams (Wiskott & Sejnowski, 2002). Their approach attempts to build feature representations that extract

---

Code and video examples can be found at: `https://coxlab.github.io/prednet/`

slowly-varying parameters, such as object identity, from parameters that produce fast changes in the image, such as movement of the object. While approaches that rely on temporal coherence have arguably not yet yielded representations as powerful as those learned by supervised methods, they nonetheless point to the potential of learning useful representations from video (Mohabi et al., 2009; Sun et al., 2014; Goroshin et al., 2015a; Maltoni & Lomonaco, 2015; Wang & Gupta, 2015).

Here, we explore another potential principle for exploiting video for unsupervised learning: prediction of future image frames (Softky, 1996; Palm, 2012; O'Reilly et al., 2014; Goroshin et al., 2015b; Srivastava et al., 2015; Mathieu et al., 2016; Patraucean et al., 2015; Finn et al., 2016; Vondrick et al., 2016). A key insight here is that in order to be able to predict how the visual world will change over time, an agent must have at least some implicit model of object structure and the possible transformations objects can undergo. To this end, we have designed a neural network architecture, which we informally call a "PredNet," that attempts to continually predict the appearance of future video frames, using a deep, recurrent convolutional network with both bottom-up and top-down connections. Our work here builds on previous work in next-frame video prediction (Ranzato et al., 2014; Michalski et al., 2014; Srivastava et al., 2015; Mathieu et al., 2016; Lotter et al., 2015; Patraucean et al., 2015; Oh et al., 2015; Finn et al., 2016; Xue et al., 2016; Vondrick et al., 2016; Brabandere et al., 2016), but we take particular inspiration from the concept of "predictive coding" from the neuroscience literature (Rao & Ballard, 1999; Rao & Sejnowski, 2000; Lee & Mumford, 2003; Friston, 2005; Summerfield et al., 2006; Egner et al., 2010; Bastos et al., 2012; Spratling, 2012; Chalasani & Principe, 2013; Clark, 2013; O'Reilly et al., 2014; Kanai et al., 2015). Predictive coding posits that the brain is continually making predictions of incoming sensory stimuli (Rao & Ballard, 1999; Friston, 2005). Top-down (and perhaps lateral) connections convey these predictions, which are compared against actual observations to generate an error signal. The error signal is then propagated back up the hierarchy, eventually leading to an update of the predictions.

We demonstrate the effectiveness of our model for both synthetic sequences, where we have access to the underlying generative model and can investigate what the model learns, as well as natural videos. Consistent with the idea that prediction requires knowledge of object structure, we find that these networks successfully learn internal representations that are well-suited to subsequent recognition and decoding of latent object parameters (e.g. identity, view, rotation speed, etc.). We also find that our architecture can scale effectively to natural image sequences, by training using car-mounted camera videos. The network is able to successfully learn to predict both the movement of the camera and the movement of objects in the camera's view. Again supporting the notion of prediction as an unsupervised learning rule, the model's learned representation in this setting supports decoding of the current steering angle.

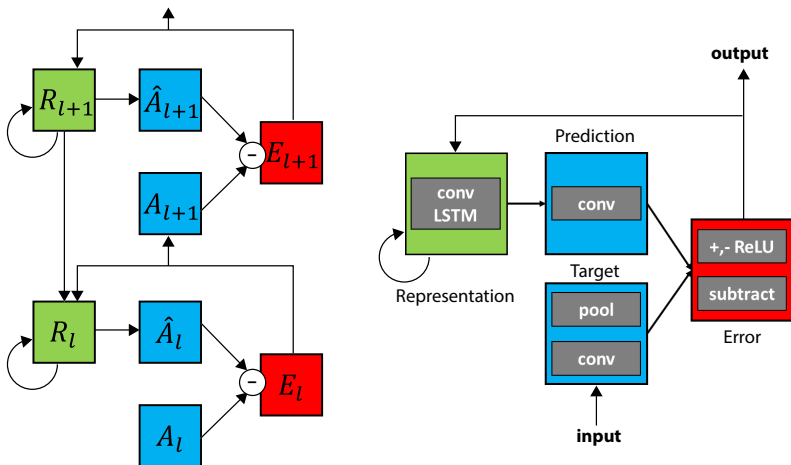

Figure 1: Predictive Coding Network (PredNet). Left: Illustration of information flow within two layers. Each layer consists of representation neurons ($R_l$), which output a layer-specific prediction at each time step ($\hat{A}_l$), which is compared against a target ($A_l$) (Bengio, 2014) to produce an error term ($E_l$), which is then propagated laterally and vertically in the network. Right: Module operations for case of video sequences.

## 2 THE PREDNET MODEL

The PredNet architecture is diagrammed in Figure 1. The network consists of a series of repeating stacked modules that attempt to make local predictions of the input to the module, which is then subtracted from the actual input and passed along to the next layer. Briefly, each module of the network consists of four basic parts: an input convolutional layer ($A_l$), a recurrent representation layer ($R_l$), a prediction layer ($\hat{A}_l$), and an error representation ($E_l$). The representation layer, $R_l$, is a recurrent convolutional network that generates a prediction, $\hat{A}_l$, of what the layer input, $A_l$, will be on the next frame. The network takes the difference between $A_l$ and $\hat{A}_l$ and outputs an error representation, $E_l$, which is split into separate rectified positive and negative error populations. The error, $E_l$, is then passed forward through a convolutional layer to become the input to the next layer ($A_{l+1}$). The recurrent prediction layer $R_l$ receives a copy of the error signal $E_l$, along with top-down input from the representation layer of the next level of the network ($R_{l+1}$). The organization of the network is such that on the first time step of operation, the "right" side of the network ($A_l$'s and $E_l$'s) is equivalent to a standard deep convolutional network. Meanwhile, the "left" side of the network (the $R_l$'s) is equivalent to a generative deconvolutional network with local recurrence at each stage. The architecture described here is inspired by that originally proposed by (Rao & Ballard, 1999), but is formulated in a modern deep learning framework and trained end-to-end using gradient descent, with a loss function implicitly embedded in the network as the firing rates of the error neurons. Our work also shares motivation with the Deep Predictive Coding Networks of Chalasani & Principe (2013); however, their framework is based upon sparse coding and a linear dynamical system with greedy layer-wise training, whereas ours is rooted in convolutional and recurrent neural networks trained with backprop.

While the architecture is general with respect to the kinds of data it models, here we focus on image sequence (video) data. Consider a sequence of images, $x_t$. The target for the lowest layer is set to the the actual sequence itself, i.e. $A_0^t = x_t \; \forall t$. The targets for higher layers, $A_l^t$ for $l > 0$, are computed by a convolution over the error units from the layer below, $E_{l-1}^t$, followed by rectified linear unit (ReLU) activation and max-pooling. For the representation neurons, we specifically use convolutional LSTM units (Hochreiter & Schmidhuber, 1997; Shi et al., 2015). In our setting, the $R_l^t$ hidden state is updated according to $R_l^{t-1}$, $E_l^{t-1}$, as well as $R_{l+1}^t$, which is first spatially upsampled (nearest-neighbor), due to the pooling present in the feedforward path. The predictions, $\hat{A}_l^t$ are made through a convolution of the $R_l^t$ stack followed by a ReLU non-linearity. For the lowest layer, $\hat{A}_l^t$ is also passed through a saturating non-linearity set at the maximum pixel value: $\text{SatLU}(x; p_{max}) := \min(p_{max}, x)$. Finally, the error response, $E_l^t$, is calculated from the difference between $\hat{A}_l^t$ and $A_l^t$ and is split into ReLU-activated positive and negative prediction errors, which are concatenated along the feature dimension. As discussed in (Rao & Ballard, 1999), although not explicit in their model, the separate error populations are analogous to the existence of on-center, off-surround and off-center, on-surround neurons early in the visual system.

The full set of update rules are listed in Equations (1) to (4). The model is trained to minimize the weighted sum of the activity of the error units. Explicitly, the training loss is formalized in Equation 5 with weighting factors by time, $\lambda_t$, and layer, $\lambda_l$, and where $n_l$ is the number of units in the $l$th layer. With error units consisting of subtraction followed by ReLU activation, the loss at each layer is equivalent to an L1 error. Although not explored here, other error unit implementations, potentially even probabilistic or adversarial (Goodfellow et al., 2014), could also be used.

$$A_l^t = \begin{cases} x_t & \text{if } l = 0 \\ \text{MaxPool}(\text{ReLU}(\text{Conv}(E_{l-1}^t))) & l > 0 \end{cases} \tag{1}$$

$$\hat{A}_l^t = \text{ReLU}(\text{Conv}(R_l^t)) \tag{2}$$

$$E_l^t = [\text{ReLU}(A_l^t - \hat{A}_l^t); \text{ReLU}(\hat{A}_l^t - A_l^t)] \tag{3}$$

$$R_l^t = \text{ConvLSTM}(E_l^{t-1}, R_l^{t-1}, \text{Upsample}(R_{l+1}^t)) \tag{4}$$

$$L_{train} = \sum_t \lambda_t \sum_l \frac{\lambda_l}{n_l} \sum_{n_l} E_l^t \tag{5}$$

---

**Algorithm 1** Calculation of PredNet states

**Require:** $x_t$
1: $A_0^t \leftarrow x_t$
2: $E_l^0, R_l^0 \leftarrow 0$
3: **for** $t = 1$ **to** $T$ **do**
4: **for** $l = L$ **to** $0$ **do** ▷ Update $R_l^t$ states
5: **if** $l = L$ **then**
6: $R_L^t = \text{CONVLSTM}(E_L^{t-1}, R_L^{t-1})$
7: **else**
8: $R_l^t = \text{CONVLSTM}(E_l^{t-1}, R_l^{t-1}, \text{UPSAMPLE}(R_{l+1}^t))$
9: **for** $l = 0$ **to** $L$ **do** ▷ Update $\hat{A}_l^t, A_l^t, E_l^t$ states
10: **if** $l = 0$ **then**
11: $\hat{A}_0^t = \text{SATLU}(\text{RELU}(\text{CONV}(R_0^t)))$
12: **else**
13: $\hat{A}_l^t = \text{RELU}(\text{CONV}(R_l^t))$
14: $E_l^t = [\text{RELU}(A_l^t - \hat{A}_l^t); \text{RELU}(\hat{A}_l^t - A_t^l)]$
15: **if** $l < L$ **then**
16: $A_{l+1}^t = \text{MAXPOOL}(\text{CONV}(E_t^l))$

---

The order in which each unit in the model is updated must also be specified, and our implementation is described in Algorithm 1. Updating of states occurs through two passes: a top-down pass where the $R_l^t$ states are computed, and then a forward pass to calculate the predictions, errors, and higher level targets. A last detail of note is that $R_l$ and $E_l$ are initialized to zero, which, due to the convolutional nature of the network, means that the initial prediction is spatially uniform.

## 3 EXPERIMENTS

### 3.1 RENDERED IMAGE SEQUENCES

To gain an understanding of the representations learned in the proposed framework, we first trained PredNet models using synthetic images, for which we have access to the underlying generative stimulus model and all latent parameters. We created sequences of rendered faces rotating with two degrees of freedom, along the "pan" (out-of-plane) and "roll" (in-plane) axes. The faces start at a random orientation and rotate at a random constant velocity for a total of 10 frames. A different face was sampled for each sequence. The images were processed to be grayscale, with values normalized between 0 and 1, and 64x64 pixels in size. We used 16K sequences for training and 800 for both validation and testing.

Predictions generated by a PredNet model are shown in Figure 2. The model is able to accumulate information over time to make accurate predictions of future frames. Since the representation neurons are initialized to zero, the prediction at the first time step is uniform. On the second time step, with no motion information yet, the prediction is a blurry reconstruction of the first time step. After further iterations, the model adapts to the underlying dynamics to generate predictions that closely match the incoming frame.

For choosing the hyperparameters of the model, we performed a random search and chose the model that had the lowest L1 error in frame prediction averaged over time steps 2-10 on a validation set. Given this selection criteria, the best performing models tended to have a loss solely concentrated at the lowest layer (i.e. $\lambda_0 = 1$, $\lambda_{l>0} = 0$), which is the case for the model shown. Using an equal loss at each layer considerably degraded predictions, but enforcing a moderate loss on upper layers that was one magnitude smaller than the lowest layer (i.e. $\lambda_0 = 1$, $\lambda_{l>0} = 0.1$) led to only slightly worse predictions, as illustrated in Figure 9 in the Appendix. In all cases, the time loss weight, $\lambda_t$, was set to zero for the first time step and then one for all time steps after. As for the remaining hyperparameters, the model shown has 5 layers with 3x3 filter sizes for all convolutions, max-pooling of stride 2, and number of channels per layer, for both $A_l$ and $R_l$ units, of $(1, 32, 64, 128, 256)$. Model weights were optimized using the Adam algorithm (Kingma & Ba, 2014).

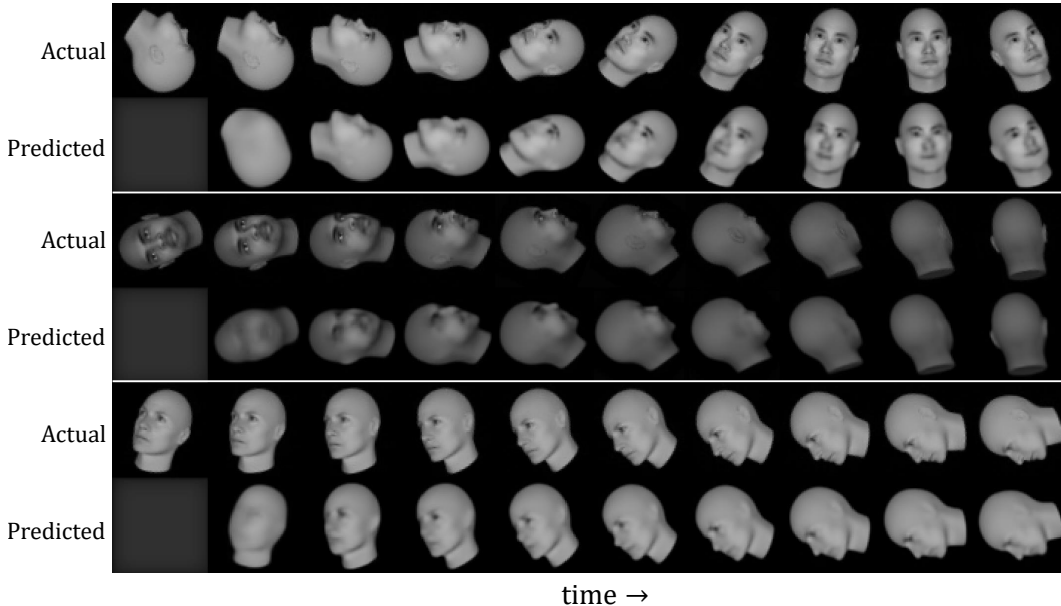

time →

Figure 2: PredNet next-frame predictions for sequences of rendered faces rotating with two degrees of freedom. Faces shown were not seen during training.

Quantitative evaluation of generative models is a difficult, unsolved problem (Theis et al., 2016), but here we report prediction error in terms of mean-squared error (MSE) and the Structural Similarity Index Measure (SSIM) (Wang et al., 2004). SSIM is designed to be more correlated with perceptual judgments, and ranges from $-1$ and $1$, with a larger score indicating greater similarity. We compare the PredNet to the trivial solution of copying the last

Table 1: Evaluation of next-frame predictions on Rotating Faces Dataset (test set).

|  | MSE | SSIM |
|---|---|---|
| PredNet $L_0$ | **0.0152** | **0.937** |
| PredNet $L_{all}$ | 0.0157 | 0.921 |
| CNN-LSTM Enc.-Dec. | 0.0180 | 0.907 |
| Copy Last Frame | 0.125 | 0.631 |

frame, as well as a control model that shares the overall architecture and training scheme of the PredNet, but that sends forward the layer-wise activations ($A_l$) rather than the errors ($E_l$). This model thus takes the form of a more traditional encoder-decoder pair, with a CNN encoder that has lateral skip connections to a convolutional LSTM decoder. The performance of all models on the rotating faces dataset is summarized in Table 1, where the scores were calculated as an average over all predictions after the first frame. We report results for the PredNet model trained with loss only on the lowest layer, denoted as PredNet $L_0$, as well as the model trained with an $0.1$ weight on upper layers, denoted as PredNet $L_{all}$. Both PredNet models outperformed the baselines on both measures, with the $L_0$ model slightly outperforming $L_{all}$, as expected for evaluating the pixel-level predictions.

Synthetic sequences were chosen as the initial training set in order to better understand what is learned in different layers of the model, specifically with respect to the underlying generative model (Kulkarni et al., 2015). The rotating faces were generated using the FaceGen software package (Singular Inversions, Inc.), which internally generates 3D face meshes by a principal component analysis in "face space", derived from a corpus of 3D face scans. Thus, the latent parameters of the image sequences used here consist of the initial pan and roll angles, the pan and roll velocities, and the principal component (PC) values, which control the "identity" of the face. To understand the information contained in the trained models, we decoded the latent parameters from the representation neurons ($R_l$) in different layers, using a ridge regression. The $R_l$ states were taken at the earliest possible informative time steps, which, in the our notation, are the second and third steps, respectively, for the static and dynamic parameters. The regression was trained using $4K$ sequences with $500$ for validation and $1K$ for testing. For a baseline comparison of the information implicitly embedded in the network architecture, we compare to the decoding accuracies of an untrained network with random initial weights. Note that in this randomly initialized case, we still expect above-chance decoding performance, given past theoretical and empirical work with random networks (Pinto et al., 2009; Jarrett et al., 2009; Saxe et al., 2010).

Latent variable decoding accuracies of the pan and roll velocities, pan initial angle, and first PC are shown in the left panel of Figure 3. There are several interesting patterns. First, the trained models learn a representation that generally permits a better linear decoding of the underlying latent factors than the randomly initialized model, with the most striking difference in terms of the the pan rotation speed ($\alpha_{pan}$). Second, the most notable difference between the $L_{all}$ and $L_0$ versions occurs with the first principle component, where the model trained with loss on all layers has a higher decoding accuracy than the model trained with loss only on the lowest layer.

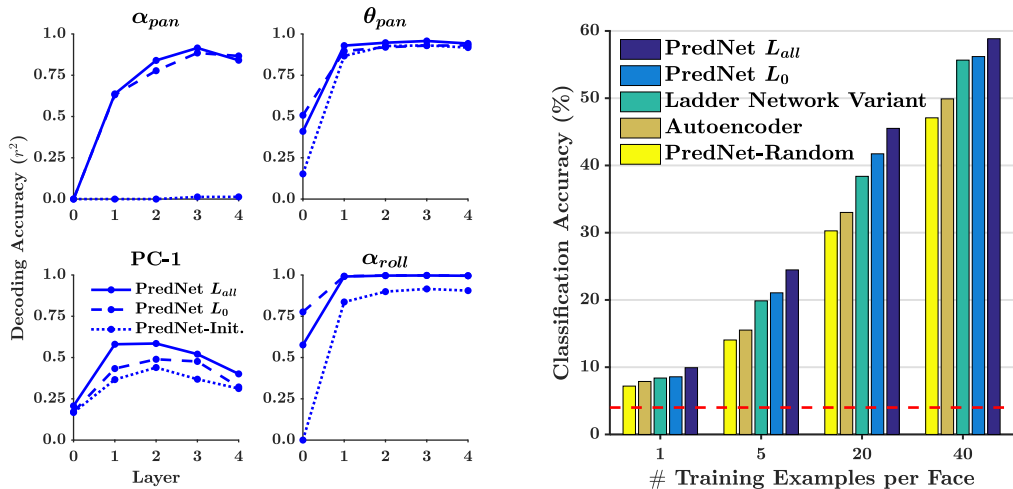

Figure 3: Information contained in PredNet representation for rotating faces sequences. Left: Decoding of latent variables using a ridge regression ($\alpha_{pan}$: pan (out-of-frame) angular velocity, $\theta_{pan}$: pan angle, PC-1: first principal component of face, $\alpha_{roll}$: roll (in-frame) angular velocity). Right: Orientation-invariant classification of static faces.

The latent variable decoding analysis suggests that the model learns a representation that may generalize well to other tasks for which it was not explicitly trained. To investigate this further, we assessed the models in a classification task from single, static images. We created a dataset of 25 previously unseen FaceGen faces at 7 pan angles, equally spaced between $\left[-\frac{\pi}{2}, \frac{\pi}{2}\right]$, and 8 roll angles, equally spaced between $[0, 2\pi)$. There were therefore $7 \cdot 8 = 56$ orientations per identity, which were tested in a cross-validated fashion. A linear SVM to decode face identity was fit on a model's representation of a random subset of orientations and then tested on the remaining angles. For each size of the SVM training set, ranging from 1-40 orientations per face, 50 different random splits were generated, with results averaged over the splits.

For the static face classification task, we compare the PredNets to a standard autoencoder and a variant of the Ladder Network (Valpola, 2015; Rasmus et al., 2015). Both models were constructed to have the same number of layers and channel sizes as the PredNets, as well as a similar alternating convolution/max-pooling, then upsampling/convolution scheme. As both networks are autoencoders, they were trained with a reconstruction loss, with a dataset consisting of all of the individual frames from the sequences used to train the PredNets. For the Ladder Network, which is a denoising autoencoder with lateral skip connections, one must also choose a noise parameter, as well as the relative weights of each layer in the total cost. We tested noise levels ranging from 0 to 0.5 in increments of 0.1, with loss weights either evenly distributed across layers, solely concentrated at the pixel layer, or 1 at the bottom layer and 0.1 at upper layers (analogous to the PredNet $L_{all}$ model). Shown is the model that performed best for classification, which consisted of 0.4 noise and only pixel weighting. Lastly, as in our architecture, the Ladder Network has lateral and top-down streams that are combined by a combinator function. Inspired by (Pezeshki et al., 2015), where a learnable MLP improved results, and to be consistent in comparing to the PredNet, we used a purely convolutional combinator. Given the distributed representation in both networks, we decoded from a concatenation of the feature representations at all layers, except the pixel layer. For the PredNets, the representation units were used and features were extracted after processing one input frame.

Face classification accuracies using the representations learned by the $L_0$ and $L_{all}$ PredNets, a standard autoencoder, and a Ladder Network variant are shown in the right panel of Figure 3. Both PredNets compare favorably to the other models at all sizes of the training set, suggesting they learn a representation that is relatively tolerant to object transformations. Similar to the decoding accuracy of the first principle component, the PredNet $L_{all}$ model actually outperformed the $L_0$ variant. Altogether, these results suggest that predictive training with the PredNet can be a viable alternative to other models trained with a more traditional reconstructive or denoising loss, and that the relative layer loss weightings ($\lambda_l$'s) may be important for the particular task at hand.

## 3.2 NATURAL IMAGE SEQUENCES

We next sought to test the PredNet architecture on complex, real-world sequences. As a testbed, we chose car-mounted camera videos, since these videos span across a wide range of settings and are characterized by rich temporal dynamics, including both self-motion of the vehicle and the motion of other objects in the scene (Agrawal et al., 2015). Models were trained using the raw videos from the KITTI dataset (Geiger et al., 2013), which were captured by a roof-mounted camera on a car driving around an urban environment in Germany. Sequences of 10 frames were sampled from the "City", "Residential", and "Road" categories, with 57 recording sessions used for training and 4 used for validation. Frames were center-cropped and downsampled to 128x160 pixels. In total, the training set consisted of roughly 41K frames.

A random hyperparameter search, with model selection based on the validation set, resulted in a 4 layer model with 3x3 convolutions and layer channel sizes of $(3, 48, 96, 192)$. Models were again trained with Adam (Kingma & Ba, 2014) using a loss either solely computed on the lowest layer ($L_0$) or with a weight of 1 on the lowest layer and 0.1 on the upper layers ($L_{all}$). Adam parameters were initially set to their default values ($\alpha = 0.001$, $\beta_1 = 0.9$, $\beta_2 = 0.999$) with the learning rate, $\alpha$, decreasing by a factor of 10 halfway through training. To assess that the network had indeed learned a robust representation, we tested on the CalTech Pedestrian dataset (Dollár et al., 2009), which consists of videos from a dashboard-mounted camera on a vehicle driving around Los Angeles. Testing sequences were made to match the frame rate of the KITTI dataset and again cropped to 128x160 pixels. Quantitative evaluation was performed on the entire CalTech test partition, split into sequences of 10 frames.

Sample PredNet predictions (for the $L_0$ model) on the CalTech Pedestrian dataset are shown in Figure 4, and example videos can be found at `https://coxlab.github.io/prednet/`. The model is able to make fairly accurate predictions in a wide range of scenarios. In the top sequence of Fig. 4, a car is passing in the opposite direction, and the model, while not perfect, is able to predict its trajectory, as well as fill in the ground it leaves behind. Similarly in Sequence 3, the model is able to predict the motion of a vehicle completing a left turn. Sequences 2 and 5 illustrate that the PredNet can judge its own movement, as it predicts the appearance of shadows and a stationary vehicle as they approach. The model makes reasonable predictions even in difficult scenarios, such as when the camera-mounted vehicle is turning. In Sequence 4, the model predicts the position of a tree, as the vehicle turns onto a road. The turning sequences also further illustrate the model's ability to "fill-in", as it is able to extrapolate sky and tree textures as unseen regions come into view. As an additional control, we show a sequence at the bottom of Fig. 4, where the input has been temporally scrambled. In this case, the model generates blurry frames, which mostly just resemble the previous frame. Finally, although the PredNet shown here was trained to predict one frame ahead, it is also possible to predict multiple frames into the future, by feeding back predictions as the inputs and recursively iterating. We explore this in Appendix 5.3.

Quantitatively, the PredNet models again outperformed the CNN-LSTM Encoder-Decoder. To ensure that the difference in performance was not simply because of the choice of hyperparameters, we trained models with four other sets of hyperparameters, which were sampled from the initial random search over the number of layers, filter sizes, and number of filters per layer. For each of the four additional sets, the PredNet $L_0$ had the best performance, with an average error reduction of 14.7% and 14.9% for MSE and SSIM,

Table 2: Evaluation of Next-Frame Predictions on CalTech Pedestrian Dataset.

|  | MSE | SSIM |
|---|---|---|
| PredNet $L_0$ | $\mathbf{3.13 \times 10^{-3}}$ | **0.884** |
| PredNet $L_{all}$ | $3.33 \times 10^{-3}$ | 0.875 |
| CNN-LSTM Enc.-Dec. | $3.67 \times 10^{-3}$ | 0.865 |
| Copy Last Frame | $7.95 \times 10^{-3}$ | 0.762 |

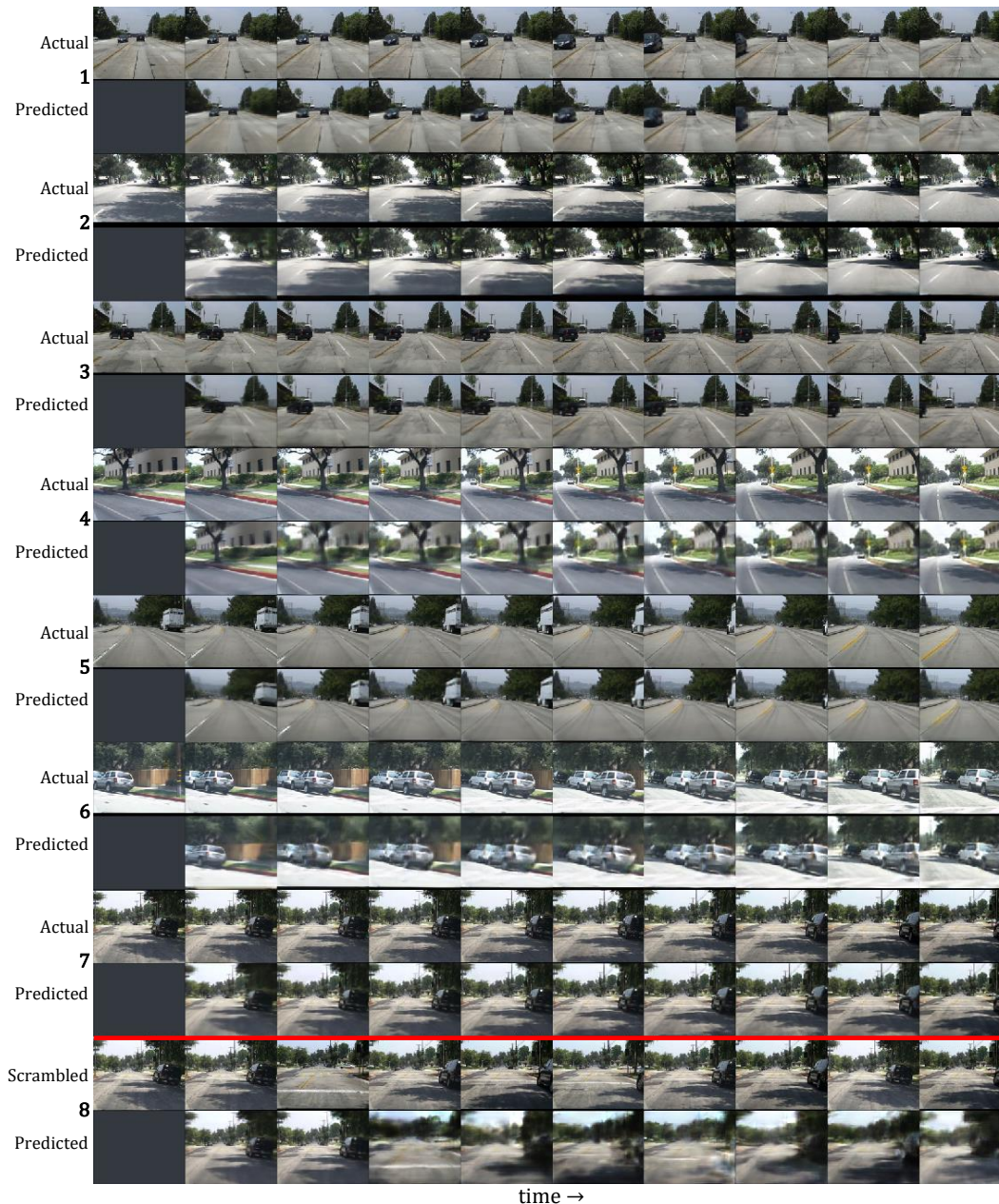

Figure 4: PredNet predictions for car-cam videos. The first rows contain ground truth and the second rows contain predictions. The sequence below the red line was temporally scrambled. The model was trained on the KITTI dataset and sequences shown are from the CalTech Pedestrian dataset.

respectively, compared to the CNN-LSTM Encoder-Decoder. More details, as well as a thorough investigation of systematically simplified models on the continuum between the PredNet and the CNN-LSTM Encoder-Decoder can be found in Appendix 5.1. Briefly, the elementwise subtraction operation in the PredNet seems to be beneficial, and the nonlinearity of positive/negative splitting also adds modest improvements. Finally, while these experiments measure the benefits of each component of our model, we also directly compare against recent work in a similar car-cam setting, by reporting results on a 64x64 pixel, grayscale car-cam dataset released by Brabandere et al. (2016). Our PredNet model outperforms the model by Brabandere et al. (2016) by 29%. Details can be found in Appendix 5.2. Also in Appendix 5.2, we present results for the Human3.6M (Ionescu et al., 2014) dataset, as reported by Finn et al. (2016). Without re-optimizing hyperparameters, our

model underperforms the concurrently developed DNA model by Finn et al. (2016), but outperforms the model by Mathieu et al. (2016).

To test the implicit encoding of latent parameters in the car-cam setting, we used the internal representation in the PredNet to estimate the car's steering angle (Bojarski et al., 2016; Biasini et al., 2016). We used a dataset released by Comma.ai (Biasini et al., 2016) consisting of 11 videos totaling about 7 hours of mostly highway driving. We first trained networks for next-frame prediction and then fit a linear fully-connected layer on the learned representation to estimate the steering angle, using a MSE loss. We again concatenate the $R_l$ representation at all layers, but first spatially average pool lower layers to match the spatial size of the upper layer, in order to reduce dimensionality. Steering angle estimation results, using the representation on the $10^{\text{th}}$ time step, are shown in Figure 5. Given just 1K labeled training examples, a simple linear readout on the PredNet $L_0$ representation explains 74% of the variance in the steering angle and outperforms the CNN-LSTM Enc.-Dec. by 35%. With 25K labeled training examples, the PredNet $L_0$ has a MSE (in *degrees*$^2$) of 2.14. As a point of reference, a CNN model designed to predict the steering angle (Biasini et al., 2016), albeit from a single frame instead of multiple frames, achieve a MSE of ~4 when trained end-to-end using 396K labeled training examples. Details of this analysis can be found in Appendix 8. Interestingly, in this task, the PredNet $L_{all}$ model actually underperformed the $L_0$ model and slightly underperformed the CNN-LSTM Enc.-Dec, again suggesting that the $\lambda_l$ parameter can affect the representation learned, and different values may be preferable in different end tasks. Nonetheless, the readout from the $L_{all}$ model still explained a substantial proportion of the steering angle variance and strongly outperformed the random initial weights. Overall, this analysis again demonstrates that a representation learned through prediction, and particularly with the PredNet model with appropriate hyperparameters, can contain useful information about underlying latent parameters.

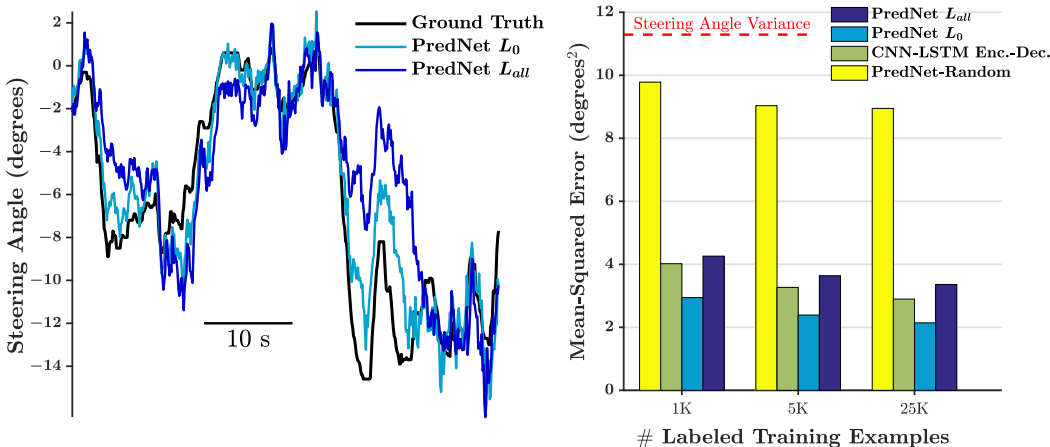

Figure 5: Steering angle estimation accuracy on the Comma.ai dataset (Biasini et al., 2016). Left: Example steering angle curve with model estimations for a segment in the test set. Decoding was performed using a fully-connected readout on the PredNet representation trained with 25K labeled training examples. PredNet representation was trained for next-frame prediction on Comma.ai training set. Right: Mean-squared error of steering angle estimation.

# 4 DISCUSSION

Above, we have demonstrated a predictive coding inspired architecture that is able to predict future frames in both synthetic and natural image sequences. Importantly, we have shown that learning to predict how an object or scene will move in a future frame confers advantages in decoding latent parameters (such as viewing angle) that give rise to an object's appearance, and can improve recognition performance. More generally, we argue that prediction can serve as a powerful unsupervised learning signal, since accurately predicting future frames requires at least an implicit model of the objects that make up the scene and how they are allowed to move. Developing a deeper understanding of the nature of the representations learned by the networks, and extending the architecture, by, for instance, allowing sampling, are important future directions.

ACKNOWLEDGMENTS

We would like to thank Rasmus Berg Palm for fruitful discussions and early brainstorming. We would also like to thank the developers of Keras (Chollet, 2016). This work was supported by IARPA (contract D16PC00002), the National Science Foundation (NSF IIS 1409097), and the Center for Brains, Minds and Machines (CBMM, NSF STC award CCF-1231216).

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

# 5 APPENDIX

## 5.1 ADDITIONAL CONTROL MODELS

Table 3 contains results for additional variations of the PredNet and CNN-LSTM Encoder-Decoder evaluated on the CalTech Pedestrian Dataset after being trained on KITTI. We evaluate the models in terms of pixel prediction, thus using the PredNet model trained with loss only on the lowest layer (PredNet $L_0$) as the base model. In addition to mean-squared error (MSE) and the Structural Similarity Index Measure (SSIM), we include calculations of the Peak Signal-To-Noise Ratio (PSNR). For each model, we evaluate it with the original set of hyperparameters (controlling the number of layers, filter sizes, and number of filters per layer), as well as with the four additional sets of hyperparameters that were randomly sampled from the initial random search (see main text for more details). Below is an explanation of the additional control models:

- **PredNet (no E split)**: PredNet model except the error responses ($E_l$) are simply linear ($\hat{A}_l - A_l$) instead of being split into positive and negative rectifications.
- **CNN-LSTM Enc.-Dec. (2x $A_l$ filts)**: CNN-LSTM Encoder-Decoder model ($A_l$'s are passed instead of $E_l$'s) except the number of filters in $A_l$ is doubled. This controls for the total number of filters in the model compared to the PredNet, since the PredNet has filters to produce $\hat{A}_l$ at each layer, which is integrated into the model's feedforward response.
- **CNN-LSTM Enc.-Dec. (except pass $E_0$)**: CNN-LSTM Encoder-Decoder model except the error is passed at the lowest layer. All remaining layers pass the activations $A_l$. With training loss taken at only the lowest layer, this variation allows us to determine if the "prediction" subtraction operation in upper layers, which is essentially unconstrained and learnable in the $L_0$ case, aids in the model's performance.
- **CNN-LSTM Enc.-Dec. (+/- split)**: CNN-LSTM Encoder-Decoder model except the activations $A_l$ are split into positive and negative populations before being passed to other layers in the network. This isolates the effect of the additional nonlinearity introduced by this procedure.

Table 3: Quantitative evaluation of additional controls for next-frame prediction in CalTech Pedestrian Dataset after training on KITTI. First number indicates score with original hyperparameters. Number in parenthesis indicates score averaged over total of five different hyperparameters.

|  | MSE (x $10^{-3}$) | PSNR | SSIM |
| --- | --- | --- | --- |
| PredNet | **3.13 (3.33)** | **25.8 (25.5)** | **0.884 (0.878)** |
| PredNet (no $E_l$ split) | 3.20 (3.37) | 25.6 (25.4) | 0.883 (0.878) |
| CNN-LSTM Enc.-Dec. | 3.67 (3.91) | 25.0 (24.6) | 0.865 (0.856) |
| CNN-LSTM Enc.-Dec. (2x $A_l$ filts) | 3.82 (3.97) | 24.8 (24.6) | 0.857 (0.853) |
| CNN-LSTM Enc.-Dec. (except pass $E_0$) | 3.41 (3.61) | 25.4 (25.1) | 0.873 (0.866) |
| CNN-LSTM Enc.-Dec. (+/- split) | 3.71 (3.84) | 24.9 (24.7) | 0.861 (0.857) |
| Copy Last Frame | 7.95 | 20.0 | 0.762 |

Equalizing the number of filters in the CNN-LSTM Encoder-Decoder (2x $A_l$ filts) cannot account for its performance difference with the PredNet, and actually leads to overfitting and a decrease in performance. Passing the error at the lowest layer ($E_0$) in the CNN-LSTM Enc.-Dec. improves performance, but still does not match the PredNet, where errors are passed at all layers. Finally, splitting the activations $A_l$ into positive and negative populations in the CNN-LSTM Enc.-Dec. does not help, but the PredNet with linear error activation ("no $E_l$ split") performs slightly worse than the original split version. Together, these results suggest that the PredNet's error passing operation can lead to improvements in next-frame prediction performance.

## 5.2 COMPARING AGAINST OTHER MODELS

While our main comparison in the text was a control model that isolates the effects of the more unique components in the PredNet, here we directly compare against other published models. We report results on a 64x64 pixel, grayscale car-cam dataset and the Human3.6M dataset (Ionescu et al., 2014) to compare against the two concurrently developed models by Brabandere et al. (2016)

and Finn et al. (2016), respectively. For both comparisons, we use a model with the same hyperparameters (# of layers, # of filters, etc.) of the PredNet $L_0$ model trained on KITTI, but train from scratch on the new datasets. The only modification we make is to train using an L2 loss instead of the effective L1 loss, since both models train with an L2 loss and report results using L2-based metrics (MSE for Brabandere et al. (2016) and PSNR for Finn et al. (2016)). That is, we keep the original PredNet model intact but directly optimize using MSE between actual and predicted frames. We measure next-frame prediction performance after inputting 3 frames and 10 frames, respectively, for the 64x64 car-cam and Human3.6M datasets, to be consistent with the published works. We also include the results using a feedforward multi-scale network, similar to the model of Mathieu et al. (2016), on Human3.6M, as reported by Finn et al. (2016).

Table 4: Evaluation of Next-Frame Predictions on 64x64 Car-Cam Dataset.

| | MSE (per-pixel) |
|---|---|
| DFN (Brabandere et al., 2016) | $1.71 \times 10^{-3}$ |
| PredNet | $\mathbf{1.16 \times 10^{-3}}$ |
| Copy Last Frame | $3.58 \times 10^{-3}$ |

Table 5: Evaluation of Next-Frame Predictions on Human3.6M

| | PSNR |
|---|---|
| DNA (Finn et al., 2016) | **42.1** |
| PredNet | 38.9 |
| FF multi-scale (Mathieu et al., 2016) | 26.7 |
| Copy Last Frame | 32.0 |

On a dataset similar to KITTI, our model outperforms the model proposed by Brabandere et al. (2016). On Human3.6M, our model outperforms a model similar to (Mathieu et al., 2016), but underperforms Finn et al. (2016), although we note we did not perform any hyperparameter optimization.

## 5.3 MULTIPLE TIME STEP PREDICTION

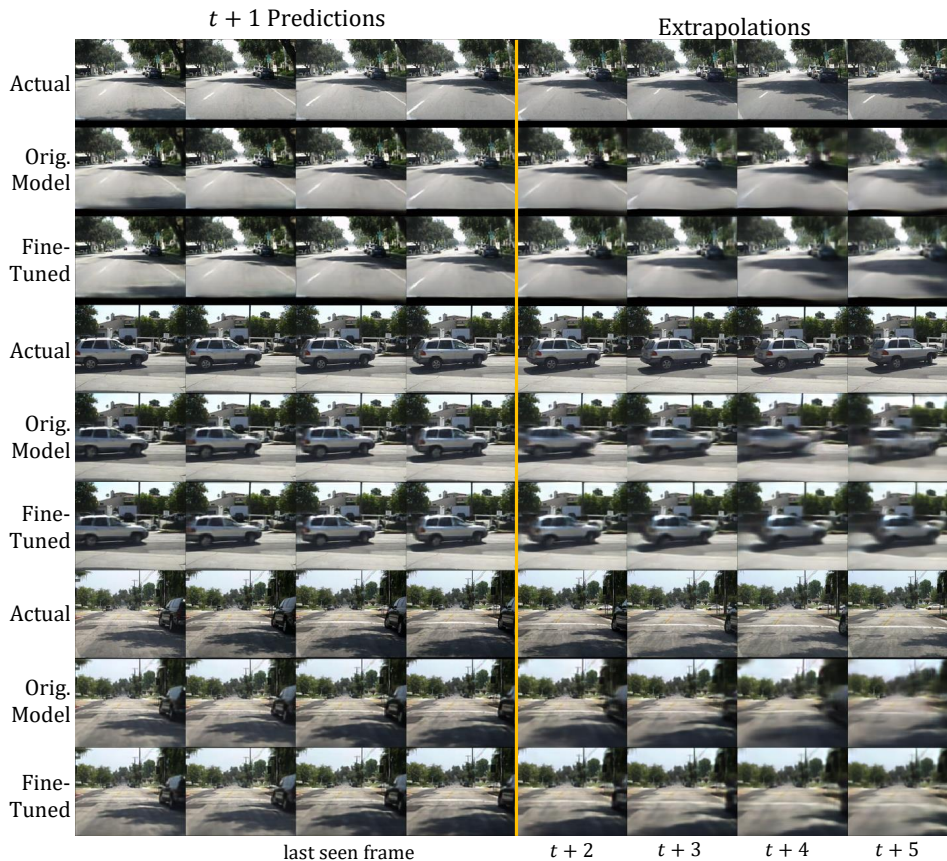

Figure 6: Extrapolation sequences generated by feeding PredNet predictions back into model. Left of the orange line: Normal $t+1$ predictions; Right: Generated by recursively using the predictions as input. First row: Ground truth sequences. Second row: Generated frames of the original model, trained to solely predict $t+1$. Third row: Model fine-tuned for extrapolation.

While the models presented here were originally trained to predict one frame ahead, they can be made to predict multiple frames by treating predictions as actual input and recursively iterating. Examples of this process are shown in Figure 6 for the PredNet $L_0$ model. Although the next frame predictions are reasonably accurate, the model naturally breaks down when extrapolating further into the future. This is not surprising since the predictions will unavoidably have different statistics than the natural images for which the model was trained to handle (Bengio et al., 2015). If we additionally train the model to process its own predictions, the model is better able to extrapolate. The third row for every sequence shows the output of the original PredNet fine-tuned for extrapolation. Starting from the trained weights, the model was trained with a loss over 15 time steps, where the actual frame was inputted for the first 10 and then the model's predictions were used as input to the network for the last 5. For the first 10 time steps, the training loss was calculated on the $E_l$ activations as usual, and for the last 5, it was calculated directly as the mean absolute error with respect to the ground truth frames. Despite eventual blurriness (which might be expected to some extent due to uncertainty), the fine-tuned model captures some key structure in its extrapolations after the tenth time step. For instance, in the first sequence, the model estimates the general shape of an upcoming shadow, despite minimal information in the last seen frame. In the second sequence, the model is able to extrapolate the motion of a car moving to the right. The reader is again encouraged to visit `https://coxlab.github.io/prednet/` to view the predictions in video form. Quantitatively, the MSE of the model's predictions stay well below the trivial solution of copying the last seen frame, as illustrated in Fig 7. The MSE increases fairly linearly from time steps 2-10, even though the model was only trained for up to $t + 5$ prediction.

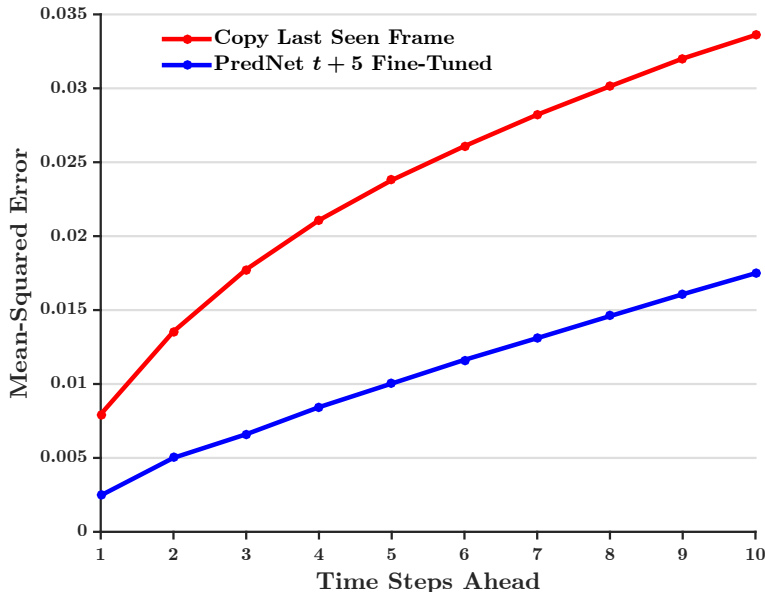

Figure 7: MSE of PredNet predictions as a function of number of time steps ahead predicted. Model was fine-tuned for up to $t + 5$ prediction.

## 5.4 ADDITIONAL STEERING ANGLE ANALYSIS

In Figure 8, we show the steering angle estimation accuracy on the Comma.ai (Biasini et al., 2016) dataset using the representation learned by the PredNet $L_0$ model, as a function of the number of frames inputted into the model. The PredNet's representation at all layers was concatenated (after spatially pooling lower layers to a common spatial resolution) and a fully-connected readout was fit using MSE. For each level of the number of training examples, we average over 10 cross-validation splits. To serve as points of reference, we include results for two static models. The first model is an autoencoder trained on single frame reconstruction with appropriately matching hyperparameters. A fully-connected layer was fit on the autoencoder's representation to estimate the steering angle in the same fashion as the PredNet. The second model is the default model in the posted Comma.ai code (Biasini et al., 2016), which is a five layer CNN. This model is trained end-to-end to estimate

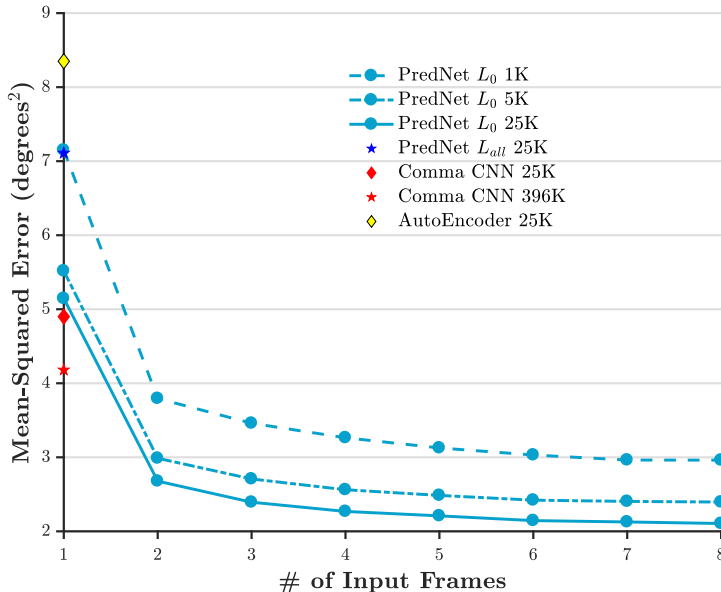

Figure 8: Steering angle estimation accuracy as a function of the number of input frames.

the steering angle given the current frame as input, with a MSE loss. In addition to 25K examples, we trained a version using all of the frames in the Comma dataset (~396K). For all models, the final weights were chosen at the minimum validation error during training. Given the relatively small number of videos in the dataset compared to the average duration of each video, we used 5% of each video for validation and testing, chosen as a random continuous chunk, and discarded the 10 frames before and after the chosen segments from the training set.

As illustrated in Figure 8, the PredNet's performance gets better over time, as one might expect, as the model is able to accumulate more information. Interestingly, it performs reasonably well after just one time step, in a regime that is orthogonal to the training procedure of the PredNet where there are no dynamics. Altogether, these results again point to the usefulness of the model in learning underlying latent parameters.

## 5.5 PREDNET $L_{all}$ NEXT-FRAME PREDICTIONS

Figures 9 and 10 compare next-frame predictions by the PredNet $L_{all}$ model, trained with a prediction loss on all layers ($\lambda_0 = 1$, $\lambda_{l>0} = 0.1$), and the PredNet $L_0$ model, trained with a loss only on the lowest layer. At first glance, the difference in predictions seem fairly minor, and indeed, in terms of MSE, the $L_{all}$ model only underperformed the $L_0$ version by 3% and 6%, respectively, for the rotating faces and CalTech Pedestrian datasets. Upon careful inspection, however, it is apparent that the $L_{all}$ predictions lack some of the finer details of the $L_0$ predictions and are more blurry in regions of high variance. For instance, with the rotating faces, the facial features are less defined and with CalTech, details of approaching shadows and cars are less precise.

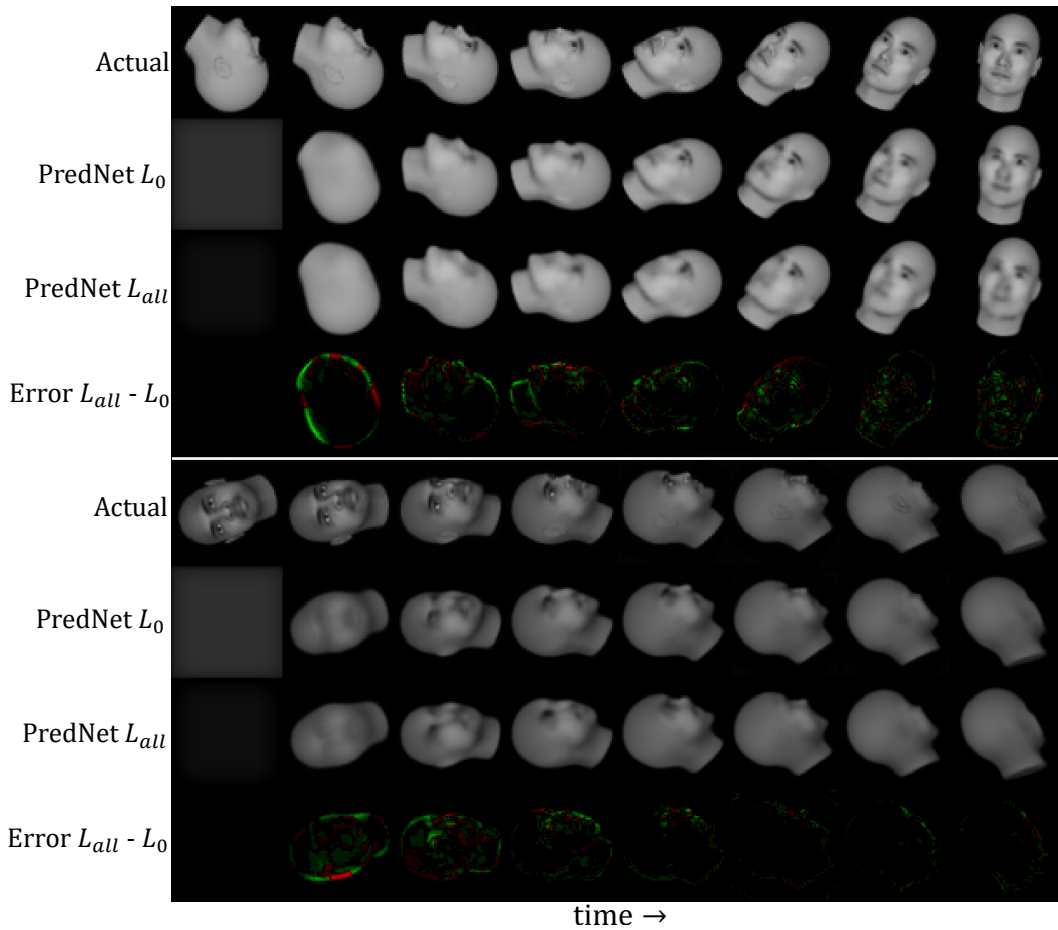

Actual

PredNet $L_0$

PredNet $L_{all}$

Error $L_{all}$ - $L_0$

Actual

PredNet $L_0$

PredNet $L_{all}$

Error $L_{all}$ - $L_0$

time →

Figure 9: Next-frame predictions of PredNet $L_{all}$ model on the rotating faces dataset and comparison to $L_0$ version. The "Error $L_{all}-L_0$" visualization shows where the pixel error was smaller for the $L_0$ model than the $L_{all}$ model. Green regions correspond to where $L_0$ was better and red corresponds to where $L_{all}$ was better.

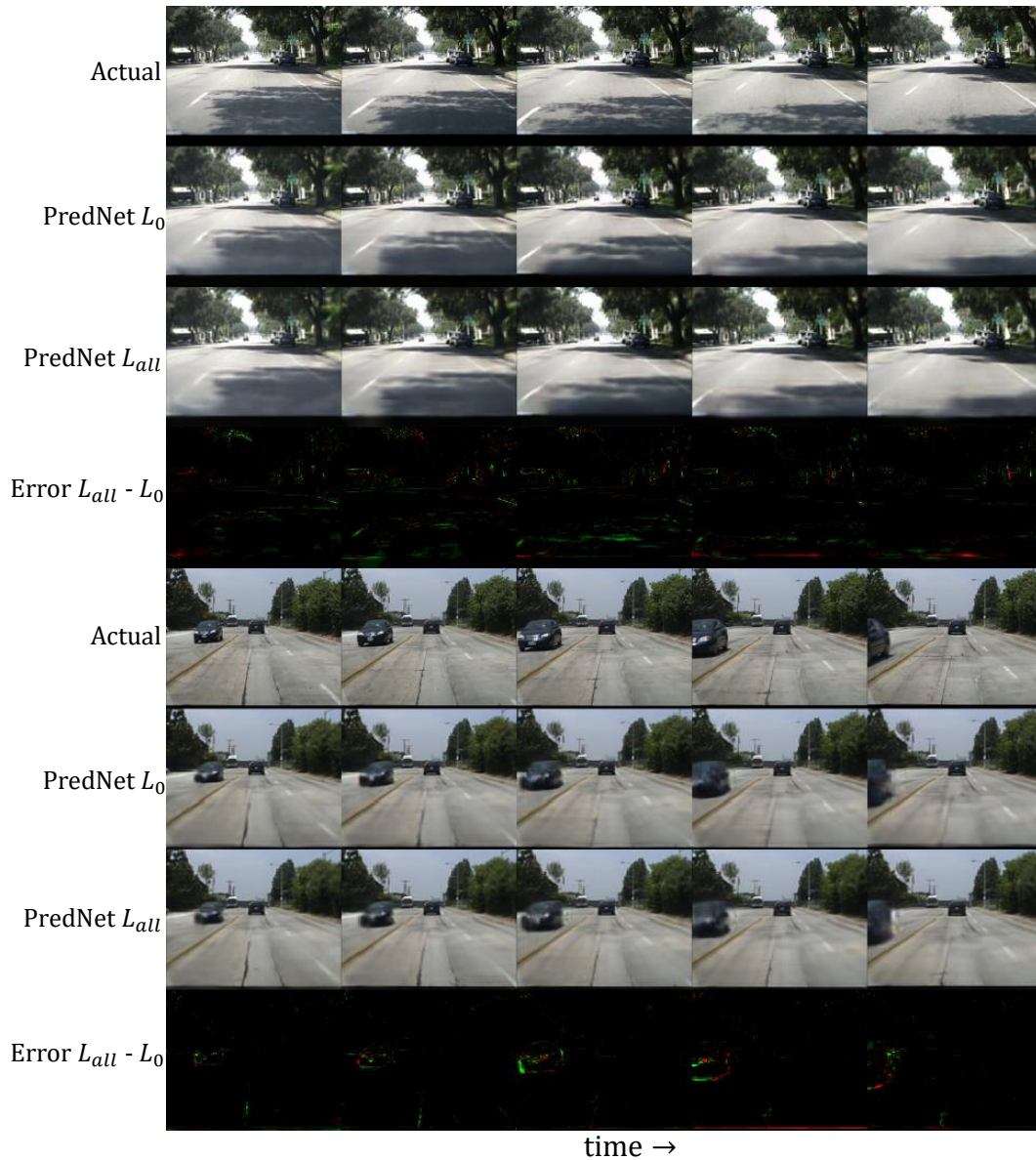

Figure 10: Next-frame predictions of PredNet $L_{all}$ model on the CalTech Pedestrian dataset and comparison to $L_0$ version. The "Error $L_{all} - L_0$" visualization shows where the pixel error was smaller for the $L_0$ model than the $L_{all}$ model. Green regions correspond to where $L_0$ was better and red corresponds to where $L_{all}$ was better.

