# Peer review of "Deep Predictive Coding Networks for Video Prediction and Unsupervised Learning"

_ICLR 2017 — accepted_

[Public Comment · (anonymous) · 02 Dec 2016]
**Questions**

What's the difference between this work and "Deep Predictive Coding Networks" (ref [4]).
Please explain it clearly in the text.

Why not comparing the performance of prednet with the previous work of authors(ref[25])?

The improvement in MSE of prednet over previous frame prediction is 0.005. How significant is this?
Maybe you could provide images showing the difference between images in t+1 and t, and between t+1 and your prediction.
This could reveal in which regions prednet does better than previous frame prediction.

[Author Response · William Lotter · 13 Dec 2016]
**Update for all reviewers and commenters**

In response to the helpful comments and questions, we have made several changes to the manuscript:

1.  In our original manuscript, we primarily compared the PredNet to a CNN-LSTM Encoder-Decoder, which we chose because it serves as a tight control for the more novel elements of our architecture. However, we agree that it is useful to compare against other published architectures.  One reason that this isn’t a trivial task is because a standard benchmark for next frame prediction arguably has yet to be established.  Another issue is that published models are often optimized for performance on particular datasets, so evaluating competing models on KITTI/CalTech isn’t necessarily fair to those models.  Searching the very recent literature, we found that the most relevant comparison to make is probably against the DFN model by Brabandere et al. (2016), which was recently presented at NIPS and was developed concurrently with our work.  One of their experiments was on a 64x64 pixel, grayscale car-cam dataset.  Training our KITTI model on this dataset, we outperform their results by 29%.  To compare against another concurrently developed model, also published at NIPS 2016, we have additionally evaluated on the Human3.6M dataset (Ionescu et al., 2014).  Our model with hyperparameters optimized for KITTI underperforms the model of Finn et al. (2016), but outperforms the previous state-of-the-art model by Mathieu et al. (2016).  We have added all of these comparisons to the appendix.

2.  To make the main text more clear and concise, and to properly explain all of the necessary details, we have moved portions of the steering angle analysis to the appendix.  Our main point has been to demonstrate that our model learns a representation of important underlying factors, using other models as points of reference, so we have emphasized this.

At the reviewer’s suggestion, we have added a video clip to help illustrate the flow of information in the network:

[Official Review · AnonReviewer2 · rating 8 · confidence 4 · 16 Dec 2016]
originality 4

Learning about the physical structure and semantics of the world from video (without supervision) is a very hot area in computer vision and machine learning.
In this paper, the authors investigate how the prediction of future image frames (inherently unsupervised) can help to deduce object/s structure and it's properties (in this case single object pose, category, and steering angle, (after a supervised linear readout step))

I enjoyed reading this paper, it is clear, interesting and proposes an original network architecture (PredNet) for video frame prediction that has produced promising results on both synthetic and natural images.
Moreover, the extensive experimental evaluation and analysis the authors provide puts it on solid ground to which others can compare.

The weaknesses:
- the link to predictive coding should be better explained in the paper if it is to be used as a motivation for the prednet model.
- any idea that the proposed method is learning an implicit `model' of the `objects' that make up the `scene' is vague and far fetched, but it sounds great.

Minor comment:
Next to the number of labeled training examples (Fig.5), it would be interesting to see how much unsupervised training data was used to train your representations.

[Official Review · AnonReviewer3 · rating 8 · confidence 5 · 16 Dec 2016]
**Good paper, nice example of using the idea of feeding forward error signals.**
originality 4 · clarity 5 · meaningful comparison 2

Paper Summary
This paper proposes an unsupervised learning model in which the network
predicts what its state would look like at the next time step (at input layer
and potentially other layers).  When these states are observed, an error signal
is computed by comparing the predictions and the observations. This error
signal is fed back into the model. The authors show that this model is able to
make good predictions on a toy dataset of rotating 3D faces as well as on
natural videos. They also show that these features help perform supervised
tasks.

Strengths
- The model is an interesting embodiment of the idea of predictive coding
  implemented using a end-to-end backpropable recurrent neural network architecture.
- The idea of feeding forward an error signal is perhaps not used as widely as it could
  be, and this work shows a compelling example of using it. 
- Strong empirical results and relevant comparisons show that the model works well.
- The authors present a detailed ablative analysis of the proposed model.

Weaknesses
- The model (esp. in Fig 1) is presented as a generalized predictive model
  where next step predictions are made at each layer. However, as discovered by
running the experiments, only the predictions at the input layer are the ones
that actually matter and the optimal choice seems to be to turn off the error
signal from the higher layers. While the authors intend to address this in future
work, I think this point merits some more discussion in the current work, given
the way this model is presented.
- The network currently lacks stochasticity and does not model the future as a
  multimodal distribution (However, this is mentioned as potential future work).

Quality
The experiments are well-designed and a detailed analysis is provided
in the appendix.

Clarity
The paper is well-written and easy to follow.

Originality
Some deep models have previously been proposed that use predictive coding.
However, the proposed model is most probably novel in the way it feds back the
error signal and implements the entire model as a single differentiable
network.

Significance
This paper will be of wide interest to the growing set of researchers working
in unsupervised learning of time series. This helps draw attention to
predictive coding as an important learning paradigm.

Overall
Good paper with detailed and well-designed experiments. The idea of feeding
forward the error signal is not being used as much as it could be in our
community. This work helps to draw the community's attention to this idea.

[Official Review · AnonReviewer1 · rating 6 · confidence 3 · 19 Dec 2016]
**an interesting architecture for future prediction inspired by deep predictive coding**
originality 4 · meaningful comparison 5

An interesting architecture that accumulates and continuously corrects mistakes as you see more and more of a video sequence.

Clarity: The video you generated seems very helpful towards understanding the information flow in your network, it would be nice to link to it from the paper.

 "Our model with hyperparameters optimized for KITTI underperforms the model of Finn et al. (2016), but outperforms the previous state-of-the-art model by Mathieu et al. (2016)."

 It is not clear how different are the train and test sequences at the moment, since standard benchmarks do not really exist for video prediction and each author picks his/her favorite. Underperforming Finn et al 206 at the H3.6m Walking videos is a bit disappointing.

[Final Decision · Program Chairs · 06 Feb 2017]
**ICLR committee final decision**

This paper proposes an interesting architecture for predicting future frames of videos using end-to-end trained deep predictive coding.
  The architecture is well presented and the paper is clearly written. The experiments are extensive and convincing, include ablation analyses, and show that this architecture performs well compared to other current methods.
 Overall, this is an interesting, solid contribution.